# Cornelian Cherry (*Cornus mas* L.) Iridoid and Anthocyanin-Rich Extract Reduces Various Oxidation, Inflammation, and Adhesion Markers in a Cholesterol-Rich Diet Rabbit Model

**DOI:** 10.3390/ijms24043890

**Published:** 2023-02-15

**Authors:** Maciej Danielewski, Agnieszka Gomułkiewicz, Alicja Z. Kucharska, Agnieszka Matuszewska, Beata Nowak, Narcyz Piórecki, Małgorzata Trocha, Marta Szandruk-Bender, Paulina Jawień, Adam Szeląg, Piotr Dzięgiel, Tomasz Sozański

**Affiliations:** 1Department of Pharmacology, Wroclaw Medical University, J. Mikulicza-Radeckiego 2, 50-345 Wroclaw, Poland; 2Division of Histology and Embryology, Department of Human Morphology and Embryology, Wroclaw Medical University, T. Chalubinskiego 6a, 50-368 Wroclaw, Poland; 3Department of Fruit, Vegetable, and Plant Nutraceutical Technology, Wroclaw University of Environmental and Life Sciences, J. Chelmonskiego 37, 51-630 Wroclaw, Poland; 4Bolestraszyce Arboretum and Institute of Physiography, Bolestraszyce 130, 37-722 Wyszatyce, Poland; 5Institute of Physical Culture Sciences, Medical College, University of Rzeszow, A. Towarnickiego 3, 35-959 Rzeszow, Poland; 6Department of Biostructure and Animal Physiology, Wroclaw University of Environmental and Life Sciences, C.K. Norwida 25/27, 50-375 Wroclaw, Poland; 7Department of Physiotherapy, Wroclaw University School of Physical Education, I.J. Paderewskiego 35, 51-612 Wroclaw, Poland

**Keywords:** Cornelian cherry, iridoids, anthocyanins, atherosclerosis, atherogenesis, inflammation, oxidation, adhesion

## Abstract

Atherogenesis leads to the development of atherosclerosis, a progressive chronic disease characterized by subendothelial lipoprotein retention and endothelial impairment in the arterial wall. It develops mainly as a result of inflammation and also many other complex processes, which arise from, among others, oxidation and adhesion. Cornelian cherry (*Cornus mas* L.) fruits are abundant in iridoids and anthocyanins—compounds with potent antioxidant and anti-inflammatory activity. This study aimed to determine the effect of two different doses (10 mg and 50 mg per kg of body weight, respectively) of iridoid and anthocyanin-rich resin-purified Cornelian cherry extract on the markers that are important in the progress of inflammation, cell proliferation and adhesion, immune system cell infiltration, and atherosclerotic lesion development in a cholesterol-rich diet rabbit model. We used biobank blood and liver samples that were collected during the previous original experiment. We assessed the mRNA expression of MMP-1, MMP-9, IL-6, NOX, and VCAM-1 in the aorta, and the serum levels of VCAM-1, ICAM-1, CRP, PON-1, MCP-1, and PCT. The application of the Cornelian cherry extract at a dose of 50 mg/kg bw resulted in a significant reduction in MMP-1, IL-6, and NOX mRNA expression in the aorta and a decrease in VCAM-1, ICAM-1, PON-1, and PCT serum levels. The administration of a 10 mg/kg bw dose caused a significant decrease in serum ICAM-1, PON-1, and MCP-1. The results indicate the potential usefulness of the Cornelian cherry extract in the prevention or treatment of atherogenesis-related cardiovascular diseases, such as atherosclerosis or metabolic syndrome.

## 1. Introduction

The search for new, effective, and safe therapies is a consistent task in medicine. Despite continuous scientific progress in the diagnosis and treatment of many medical conditions, the use of more and more advanced pharmaceuticals, and the growing awareness of the pathogenesis and treatment of various diseases, there is still a need for new specifics that will meet emerging therapeutic challenges. This especially applies to conditions commonly referred to as civilization diseases, the prevalence of which is increasing, with widespread intensity in new groups of patients, e.g., they appear in a spiral number of cases in younger people, and the financial and social costs of treating these diseases are starting to burden national health systems. Such conditions can result, to a greater or lesser extent, from an incorrect lifestyle, primarily a small amount of physical activity and an incorrect (Western) diet rich in simple sugars and cholesterol—i.e., atherosclerosis, type 2 diabetes, or metabolic syndrome.

Atherosclerosis is a progressive chronic disease characterized by subendothelial lipoprotein retention and endothelial impairment in the arterial wall. It is considered one of the most important types of arteriosclerotic vascular diseases [1]. Atherosclerosis develops mainly as a result of inflammation but also from many other complex processes, which arise from, among others, oxidation and adhesion. Pro-oxidant substances, in general, may be divided into reactive oxygen species (ROS) and reactive nitrogen species (RNS). ROS contributes to several aspects of atherosclerosis development, including endothelial cell dysfunction, immune cell recruitment and activation, stimulation of inflammation, and smooth muscle cell migration and proliferation. In addition, they are involved in low-density lipoprotein (LDL-C) oxidative modification, the inactivation of nitric oxide, and the modulation of redox-sensitive signaling pathways [2].

The nicotinamide adenine dinucleotide phosphate (NADPH) oxidases (NOX) family is considered a major source of ROS in eukaryotic cells [3,4]. The correct production of NOX isoforms fulfills multiple processes that are important for normal physiology, but its up-regulation is involved in many different pathologies, including atherosclerosis, cancer, and neurodegenerative diseases [5]. NOX enzymes occur in macrophages, vascular smooth muscle cells (VSMCs), and endothelial cells, and are involved in smooth muscle cell proliferation, the production of ROS/RNS, and low-density lipoprotein oxidation [6]. The production of ROS in the blood vessels is essential for redirecting the bloodstream to the most active tissues and thus maintaining vascular homeostasis. On the other hand, the overproduction of ROS contributes to the development of cardiovascular diseases such as hypertension, atherosclerosis, diabetes, hypertrophy, and cardiac arrest [3]. VSMCs, the most predominant component cells of the blood vessel wall, play a pivotal role in regulating vascular function [7]. LDL-C oxidation and proinflammatory cytokines, such as TNF-alpha, IL-1, IL-4 or IL-6, and IFN-gamma, may induce endothelial dysfunction, migration, and the proliferation of smooth muscle cells, the emergence of foam cells, and the expression of leukocyte and monocyte adhesion molecules, primarily vascular cell adhesion molecule-1 (VCAM-1), intercellular adhesion molecule-1 (ICAM-1), and E-selectin [6]. These molecules form an integrated system that transports leukocytes and monocytes into the vascular wall, promotes their accumulation in the vascular wall intima, and contributes to the development of atherosclerotic plaques [8]. Afterward, monocytes are differentiated into macrophages through chemotactic proteins, mainly monocyte chemotactic protein-1 (MCP-1) [9].

Matrix metalloproteinases (MMPs) are part of a large family of zinc-dependent endopeptidases. The major role of MMPs is to degrade and deposit structural proteins within the extracellular matrix (ECM). The production of MMPs is stimulated mainly by oxidative stress, various growth factors, and inflammation. The up- or down-regulation of MMPs can be conducive to vascular diseases, such as hypertension or atherosclerosis, and leads to subsequent ECM remodeling. In atherosclerosis, MMPs are primarily implicated in atherosclerotic plaque formation and instability and in promoting the migration and proliferation of smooth muscle cells [10,11,12]. Moreover, the degradation of native collagen induced by MMPs augmentation results in decreased resistance of the vessels to various stresses, including mechanical damage and intimal thickening [13,14].

Human serum paraoxonase-1 (PON-1) is a calcium-dependent hydrolytic enzyme primarily synthesized in the liver. It is an antioxidant protein protecting high-density lipoproteins from oxidation [15,16]. This HDL-associated enzyme can hydrolyze oxidized LDL-cholesterol (ox-LDL), thereby exhibiting potential antiatherosclerotic properties.

One of the raw materials of natural origin that can provide compounds that are potentially effective in the prevention and treatment of atherosclerosis is Cornelian cherry (*Cornus mas* L.) fruits, which are rich in substances from the group of polyphenols, flavonoids, anthocyanins, and iridoids [17,18,19,20,21,22,23,24,25,26]. Compounds from these groups are widely known for their potential antioxidant and anti-inflammatory effects.

Our previous work proved that Cornelian cherry extract is an efficacious therapeutic agent in a cholesterol-rich diet rabbit model. We have shown, among others, the positive impact of the extract on the levels of triglycerides and adipokines, PPAR alpha and gamma expression in the aorta, and LXR expression in the liver, as well as on the reduction of the intima/media ratio in the thoracic and abdominal aorta [27]. In this study, and continuing the evaluation of this potentially effective natural remedy, we focused more on assessing the influence of the extract on the inflammatory aspect of atherosclerosis. This study aimed to determine the effect of two different doses (10 mg or 50 mg per kg of body weight, respectively) of iridoid and anthocyanin-rich resin-purified Cornelian cherry extract on the markers important in the progression of inflammation, cell proliferation, immune system cells infiltration, and atherosclerotic lesion development in a cholesterol-rich diet rabbit model. We assessed the levels of the above-mentioned markers: the mRNA expression of MMP-1, MMP-9, IL-6, NOX, and VCAM-1 in the aorta, the serum concentrations of VCAM-1, ICAM-1, PON-1, and MCP-1, and classic inflammation markers, such as C-reactive protein (CRP) and procalcitonin (PCT).

## 2. Results

We have studied the effects of the oral administration of resin-purified Cornelian cherry extract in a cholesterol-rich diet rabbit model on mRNA expression of MMP-1, MMP-9, IL-6, NOX, and VCAM-1 in the aorta. Additionally, we have indicated the levels of VCAM-1, ICAM-1, CRP, PON-1, MCP-1, and PCT in the serum.

### 2.1. Assessment of MMP-1, MMP-9, IL-6, NOX, and VCAM-1 mRNA Expression in the Aorta by Real-Time PCR

When compared to the baseline, feeding a cholesterol-rich diet caused a significant increase in the CHOL group compared to the P group in MMP-1 (*p* = 0.006), MMP-9 (*p* < 0.001), NOX (*p* = 0.023), and VCAM (*p* = 0.002) expressions. In the assessment, the IL-6 upregulation in the CHOL group was also observed but to a lesser extent (*p* = 0.056). In the case of groups receiving the extract compared to the P group, significant increases in the expression of the tested substances were observed in the following samples: MMP-1 (a relevant difference in EXT 10 group, *p* = 0.019), MMP-9 (EXT 10 and EXT 50, respectively *p* < 0.001 and *p* = 0.034) and VCAM (EXT 10 and EXT 50, respectively *p* < 0.001 and *p* = 0.033). It is worth noticing that the determination of MMP-9 and VCAM expression in the EXT 10 group showed higher levels than in the CHOL group compared to the P group, while in the remaining cases, the increases were lower than those recorded in the CHOL group.

When compared to the CHOL group, significant positive changes were observed in the EXT 50 group in three out of the five analyses. The relevant decreases in expression levels concerned MMP-1 (*p* = 0.005), IL-6 (*p* = 0.029), and NOX (*p* = 0.001). Although in the MMP-9 and VCAM-1 assay, the statistical analysis did not show significance; a noticeable favorable decrease in the EXT 50 group was observed. However, when comparing the results of the EXT 10 group with the CHOL group, no relevant differences were obtained, and the assessed levels were, as mentioned earlier, in some cases higher and, in other cases, lower than in the CHOL group. The outcomes of the mRNA expression of MMP-1, MMP-9, IL-6, NOX, and VCAM in the aorta testing are presented in Table 1 and Figure 1.

### 2.2. Assessment of VCAM-1, ICAM-1, CRP, PON-1, MCP-1, and PCT Serum Levels by ELISA

When compared to the baseline, feeding a cholesterol-rich diet caused a significant increase in the CHOL group compared to the P one in VCAM-1 (*p* < 0.001), ICAM-1 (*p* < 0.001), CRP (*p* < 0.001), PON-1 (*p* < 0.001), MCP-1 (*p* = 0.032), and PCT (*p* = 0.002) groups, i.e., in the serum levels of all the ELISA-assessed compounds. In the groups receiving the extract, the levels were also elevated compared to the P group but, in the vast majority of cases, they were lower than in the CHOL group. The only exception was the VCAM-1 concentration in the EXT 10 group, which was slightly higher than in the CHOL group. Significant increases in the serum levels of the tested substances were observed in the following samples: VCAM-1 (a relevant difference in EXT 10 and EXT 50 group, respectively *p* < 0.001 and *p* = 0.026), CRP (EXT 10 and EXT 50, respectively *p* = 0.014 and *p* = 0.005), PON-1 (EXT 10, *p* = 0.038), and PCT (EXT 10, *p* = 0.018).

When compared to the CHOL group, it was possible to notice a particularly positive effect regarding the administration of the Cornelian cherry extract on the serum levels of ICAM-1 and PON-1, where relevant differences were noted for both applied doses. The following statistical analysis results were obtained for ICAM-1: EXT 10—*p* = 0.006 and EXT 50—*p* = 0.036; for PON-1: EXT 10—*p* = 0.010 and EXT 50—*p* < 0.001. In the case of VCAM-1, MCP-1, and PCT, a significant reduction in the serum concentration was obtained in one of the extract doses, with 50 mg/kg bw twice and 10 mg/kg bw once. The specific values of the analysis results in comparison to the CHOL group are as follows: VCAM-1 (EXT 50, *p* = 0.003), MCP-1 (EXT 10, *p* = 0.019), and PCT (EXT 50, *p* = 0.043). Only in the CRP study were no significant differences observed; however, the levels of this compound in the extract groups were noticeably lower, with a greater decrease in the EXT 10 group compared to the CHOL group. The outcomes of the ELISA assessment of VCAM-1, ICAM-1, CRP, PON-1, MCP-1, and PCT serum levels are presented in Table 2 and Figure 2.

## 3. Discussion

In this study, we examined a cholesterol-fed rabbit model and the effects of administering two doses (10 mg/kg bw or 50 mg/kg bw) of resin-purified Cornelian cherry (*Cornus mas* L.) extract on the mRNA expression of MMP-1, MMP-9, IL-6, NOX, and VCAM-1 in the thoracic aorta and the serum levels of VCAM-1, ICAM-1, CRP, PON-1, MCP-1, and PCT, which represent the compounds that may loom large in the pathogenesis and progress of atherosclerotic lesions. The main conclusion of our study is that oral administration of resin-purified Cornelian cherry extract has a positive, lowering impact on various markers that are important in the progression of inflammation, cell proliferation and adhesion, immune system cell infiltration, and atherosclerotic lesion development, which may contribute to the limitation of the pathogenesis and development of atherogenesis-related cardiovascular diseases, such as atherosclerosis or metabolic syndrome. The key findings are that Cornelian cherry extract, particularly at a dose of 50 mg/kg bw, significantly reduces the mRNA expression of MMP-1, IL-6, and NOX in the aorta and decreases the serum levels of VCAM-1, ICAM-1, PON-1, MCP-1, and PCT. These findings allow us to better understand the beneficial influence of Cornelian cherry extract on the formation of atherosclerotic changes, which we reported on in our previous studies [27].

Metalloproteinases that play a key role in atherosclerotic plaque vulnerability and rupture are, i.a., MMP-1 and MMP-9 [28]. Butoi et al. [29] found that crosstalk between macrophages and smooth muscle cells precisely enhances the expression of MMP-1 and MMP-9. One of the functions of MMPs is the generation of matrices—peptides originating from the fragmentation of extracellular matrix proteins. Matrikines can substantially impact inflammatory processes and atherosclerosis development due to their ability to alter cellular migration, chemotaxis, and mitogenesis. Both MMP-1 and MMP-9 (also MMP-2, MMP-8, and MMP-12) regulate, among others, the elastin peptide (val-gly-val-ala-pro-gly matrikine) [30]. Elastin peptide may stimulate the enhancement of vascular intimal wall thickness and wall diameter [31,32]. Moreover, Hu et al. proved that MMP-1 levels are positively correlated with the occurrence of cardiovascular and cerebrovascular events in the course of carotid atherosclerosis [33].

In our study, we observed an increase in the expression of MMP-1 and MMP-9 mRNA in the aorta in the CHOL group, which is consistent with the data described above, as well as a decrease in the expression level of MMP-1 and MMP-9 in the case of the 50 mg/kg bw dose application, with a statistically relevant difference in the MMP-1 level. However, in the case of a dose of 10 mg/kg bw, there was a slight mitigation (MMP-1) and a slight enhancement (MMP-9) in expression compared to the CHOL group. It appears that these relatively small disparities can be considered as within the statistical error, and it is allowed to assume that the lower dose of the extract is too small to obtain a noticeable effect on the expression level of the enzymes tested. On the other hand, the higher dose of the extract reduces the occurrence of the factors generating an increase in the level of metalloproteinases, i.e., oxidative stress or inflammation, thus demonstrating a beneficial effect on the development and stability of atherosclerotic lesions resulting from a diet rich in cholesterol. Most of the studies conducted so far on the impact of the extract or active compounds obtained from the genus *Cornus* on metalloproteinases mainly concern MMP-3 and MMP-13 [34,35,36,37], and a diminution in the expression of these enzymes was also observed; ergo, our results are consistent with the earlier outcomes, concomitantly expanding the knowledge on this subject.

As for the adhesion molecules tested, we observed the mitigation of expression of VCAM-1 in the aorta (dose of 50 mg/kg bw) and the depletion of VCAM-1 (dose of 50 mg/kg bw) and ICAM-1 (both the dose of 10 mg/kg and 50 mg/kg bw) concentrations in the serum. Interestingly, we noted a slight augmentation of VCAM-1 expression in the aorta and serum concentration in the case of the 10 mg/kg bw dose of the extract (which additionally confirms the consistency of the results received) and the concomitantly stronger beneficial effect of this dose on ICAM-1 serum concentration. This variable impact of the 10 mg/kg bw dose on both of the adhesion molecules requires further research. However, it can be overall admitted that Cornelian cherry extract reduces the levels of VCAM-1 and ICAM-1. Similar results were obtained in a human model by Kang et al. [38] with cornuside—an iridoid glucoside isolated from the fruits of *Cornus officinalis*, where the suppression of both molecules was also observed. It is worth mentioning that, in a review of the available literature, we found only this one study assessing the effect of the raw materials obtained from the genus *Cornus* on VCAM-1 and ICAM-1.

During one of the initial phases of atherosclerosis, the recruitment of inflammatory cells from the circulation and their transendothelial migration was observed. Cellular adhesion molecules, which appear on the vascular endothelium and the circulating leukocytes in response to various inflammatory factors, play a major role in this process. Both VCAM-1 and ICAM-1 induce the potent adhesion of inflammatory cells to the surface of blood vessels. The expression of these molecules is, consequently, apparent in atherosclerotic plaques [9]. VCAM-1 expression is enhanced in endothelial cells under proinflammatory conditions, as observed in the early stages of atherosclerosis [39]. Moreover, VCAM-1 may be associated with the severity of atherosclerosis and the prediction of cardiovascular disease [8]. Similarly, although ICAM-1 is constitutively present in endothelial cells, its expression is significantly enhanced in inflammatory conditions by proinflammatory cytokines. The endothelial expression of ICAM-1 is increased in atherosclerotic tissue and animal models of atherosclerosis. Elevated levels of the circulating or soluble form of ICAM-1 are observed in various body fluids in patients with atherosclerosis, heart failure, coronary artery disease, and transplant vasculopathy. ICAM-1 directly contributes to inflammatory responses within the blood vessel wall by increasing endothelial cell activation and augmenting atherosclerotic plaque formation [40]. It also plays an essential role in the regulation of vascular permeability [41].

In our previous work [27], we reported that a diet rich in cholesterol results in an increase in PPAR-alpha and PPAR-gamma expression in the aorta, and feeding the Cornelian cherry extract lowers the levels of expression of both of these transcription factors. Currently, we proved that the expression of VCAM-1 in the aorta and serum concentrations of VCAM-1 and ICAM-1 also decreased from the use of the extract. Meanwhile, Wei et al. [1] informed that VCAM-1-targeted and PPAR-delta-agonist-loaded nanomicelles had a suppressing effect on apoptosis and the migration of oxidized LDL-C-induced human aortic vascular smooth muscle cells. It can therefore be hypothesized that the application of Cornelian cherry extract may be, to a certain extent, a comparable and valuable alternative to the above therapeutic approach, as well as a natural and feasibly cheaper treatment option for compounds obtained using synthetic or biotechnological methods, which are based on curative effects on the expression levels of receptors from the PPAR group and adhesion molecules, including VCAM-1.

In the reported study, we measured the expression levels in the aorta of IL-6, which belongs to proinflammatory and proatherogenic cytokines [42]. The administration of Cornelian cherry extract at a dose of 50 mg/kg bw caused a relevant reduction in Il-6 expression compared to the CHOL group. Interestingly, the level of IL-6 in the EXT 50 group turned out to be lower than in the positive control, i.e., the SIMV 5 group. This is a very favorable alteration as an elevated level of IL-6 is associated with cardiovascular risk [43,44]. It was proven that IL-6 signaling has a causative role in atherothrombosis [45]. A diminution in IL-6 level was also observed in at least a few other studies, but these mainly concerned *Cornus officinalis* rather than *Cornus mas* [46,47,48,49,50].

Chronic inflammation is a major contributor to age-related atherosclerosis. This may result from the associated increase in the aging of adipocytes in the bone marrow accompanied by an elevation of proinflammatory cytokines, including IL-6, and the synergy between the myeloid cells of the immune system and the vasculature via IL-6 signaling. Currently, the clinically approved agents that target this pathway (such as anti-IL-6 therapies) are already available and could reduce the risk of cardiovascular disease in elderly people [51,52]. Due to its ability to reduce the level of IL-6 expression, Cornelian cherry extract may constitute a valuable supplement for such therapy.

One of the most important reactive oxygen species that produce enzymes is the NOX group. In physiological conditions, NOX presents a low basal activity in blood vessels [53]. The augmentation of NOX-derived ROS is mainly caused by cytokines, growth factors, or high glucose levels and plays a key role in the pathogenesis of atherosclerosis [3]. In our study, we observed a significant increase in the level of NOX expression in the aorta in the CHOL group compared to the P group. Expression levels were lower in the extract groups, with the EXT 50 group noting a relevant difference compared to the CHOL group and similar to that observed in the simvastatin group. The fundamental NOX feature is the generation of reactive oxygen species in the vascular cells and in the circulating immune cells interacting with the blood vessels. While the physiological production of NOX-derived ROS contributes to the maintenance of vascular homeostasis, the hyperactivity of NOX triggers oxidative stress. In atherosclerosis, lipid peroxidation induced by activated NOX is highly deleterious and expands the free radical reactions [54,55].

Iridoids and anthocyanins possess considerable antioxidant activity. It is both indirect, through the stimulation of the antioxidant defense system, and direct, through the removal of reactive oxygen species. As for NADPH oxidase, it was shown that iridoids might modulate the AMPK/NOX4/PI3-K/AKT pathway, and anthocyanin metabolites may alter NOX activity in the endothelium [24]. The elevated level of LDL-C often observed in the course of atherosclerosis results in the increased binding of LDL-C and increased uptake by cell surface LDL receptors and may be responsible for the direct activation of NOX. NOX-generated superoxide stimulates lipid endocytosis, thus promoting plaque formation [3]. Therefore, it can be hypothesized that one of the factors that contributed to the diminished NOX expression in the extract groups was the lower LDL-C concentration when compared to the CHOL group, showed in our previous report [27]. The limited activity of NOX enzymes could have contributed to a reduction in ROS production, which may be confirmed by the favorable changes observed in the case of other markers, directly or indirectly, dependent on oxidative stress, as described in this article.

In two different studies of the active ingredients isolated from *Corni fructus* by Park et al., a positive impact on NOX levels was confirmed. 7-O-galloyl-D-sedoheptulose reduced the renal protein expression of NOX-4 and one of the subunits of NADPH oxidase-p22(phox) [56], while the administration of loganin led to a significant decrease in the expression of both NOX-4 and p22(phox) in the liver of diabetic db/db mice [57]. Comparable results were obtained among others by Lee et al. [58], Chen et al. [59], and Fang et al. [60].

Although the C-reactive protein (CRP) level enhances after various unspecific inflammatory stimuli, it is also considered one of the leading biomarkers of cardiovascular risk prediction [45]. Concomitantly, it is ambiguous to state whether CRP itself plays any causal role in atherogenesis [61]. In the CHOL group, we noted a significant increase in serum CRP level compared to the P group. In the groups fed with the extract, the CRP concentrations were slightly lower (and lesser in the EXT 10 group), but were still essentially higher compared to the control group. Thus, a certain positive trend of extract administration can be noticed, but the final result obtained in our model is relatively modest. Similarly, no significant differences in C-reactive protein levels between the studied groups were determined in the *Cornus mas* randomized controlled trial [62].

A variety of studies have been performed to investigate the clinical relevance of PON-1 in cardiovascular disease, diabetes, cancer, or neurologic diseases; however, the received data are still insufficient and, in some cases, contradictory [63]. Nonetheless, Kunutsor et al. [64] stated that there is an approximately log-linear inverse association between PON-1 activity and CVD risk, which is partly dependent on HDL-C levels. Additionally, Gautier et al. recently confirmed the association between an increase in HDL-C level and PON-1 activity and a decrease in lipid oxidation markers [65].

In the assessment of serum PON-1 level, we obtained a significant reduction in the extract-fed groups compared to the CHOL group (greater in the EXT 50). It is worth noting that, while in the case of the EXT 10 group, the measured level was still essentially higher than in the control group, the concentration of PON-1 in the EXT 50 group was comparable to that determined in the P group. In a previous study, we found that there is a noticeable dose-dependent impact of Cornelian cherry extract on cholesterol fraction levels (reduction in LDL-C and augmentation of HDL-C), although these changes, at least in the doses applied, are not statistically significant compared to the CHOL group. Nevertheless, they constitute the starting point for the assessment of serum PON-1 concentration in the present study, and the obtained results are a logical consequence of the earlier ones. A more substantial increase in HDL-C was observed in the case of a 50 mg/kg bw dose translates into the greater activity of HDL-associated PON-1, which results in a boost in LDL-C hydrolysis and a more significant reduction in its level in the serum. To our knowledge, to date, only one study of the effects of *Cornus mas* on PON-1 has been carried out and resulted in the opposite to our outcomes [66]. This supports the above-mentioned statement about the insufficient and contradictory data on PON-1 and the necessity for further research.

For the monocyte chemotactic protein-1 MCP-1, we noted lower serum levels than in the CHOL group for both doses of the extract, and interestingly, the difference was significant in the EXT 10 group. It is also worth mentioning that the MCP-1 level obtained in EXT 10 group was almost identical to that in the positive control group, SIMV 5. MCP-1, also referred to as chemokine (C-C motif) ligand 2 (CCL2), is expressed by mainly inflammatory and endothelial cells [67]. This chemokine regulates monocyte chemotaxis and T-lymphocyte differentiation and plays a crucial role in the pathogenesis of various inflammatory diseases, atherosclerosis, and cancer [68]. Its level is upregulated as a result of proinflammatory stimuli and the tissue injury associated with atherosclerotic lesions [67]. According to one of the recent studies on inflammatory atherosclerosis pathogenesis, isolated human monocytes, via trained immunity induced by the primary stimulation of ox-LDL during secondary stimulation with agonists of toll-like TLR2 and TLR4 receptors, showed an increase in the production of proinflammatory cytokines such as TNF-alpha, interleukins -6, -8, and -18 and precisely the MCP-1 chemokine [69].

The explanation for the beneficial effect of the extract on the level of MCP-1 may be due to the fact that it is rich in compounds from the anthocyanin group. Anthocyanins mediate the redox state and inflammation via various pathways. One of the most important factors is the elevation of the Nrf2 factor, the nuclear factor erythroid 2-related factor 2. It is an inducible, specific transcription factor that is a key regulator of many antioxidant responses. The Nrf2 factor contributes, among others, to a reduction in MCP-1 levels [24,70]. Other available reports on the impact of compounds or derivatives acquired from the *Cornus* genus are consistent with our results and confirm the MCP-1 lowering effect [38,49,57,71,72]. A separate matter is the fact that, in our study, the lower dose of the extract appeared to be more effective. Whether the obtained outcome is either a result of the adopted model, e.g., its length, feeding pattern, or the laboratory animals used, or results directly from the doses of the extract applied is an issue that certainly requires additional clarification.

Procalcitonin (PCT) is considered a classic marker of inflammation and is, therefore, useful in determining the pathogenesis of atherosclerosis as an inflammatory-grounded disease. We observed a significant increase in PCT levels in the CHOL group compared to the P group. In the EXT 10 group, the augmentation was also significant, but in the EXT 50 group, we noted a relevant decrease in PCT levels compared to the CHOL group. Under physiological conditions, procalcitonin is a precursor of calcitonin, i.e., a thyroid hormone involved in the regulation of calcium and phosphate metabolism in the body. However, PCT also plays a very important role in pathological conditions, being a sensitive marker of inflammation. In the course of an inflammatory reaction, procalcitonin is synthesized, i.a., by macrophages, monocytes, and hepatocytes; therefore, the evaluation of its level may be a relatively simple way of assessing the presence of inflammation in the body. Procalcitonin is suggested to be a useful marker for the assessment of inflammation related to atherosclerosis or metabolic syndrome and a risk factor for future cardiovascular events [73,74]. It is associated with several other established cardiovascular risk factors, such as C-reactive protein, hypertension, diabetes, and renal function, which limits the value of PCT as an independent cardiovascular risk predictor. PCT is also inversely correlated with HDL-C levels [75]. Additionally, procalcitonin concentration may be a useful instrument for assessing carotid wall thickening and stenosis in ischemic stroke and also the severity of coronary artery disease (CAD) [73,76].

As mentioned, the use of PCT as an individual marker for the assessment of cardiovascular diseases is limited; therefore, the determination of another classical marker of inflammation, for example, CRP, is often performed as well. Interestingly, the level of procalcitonin is usually elevated at earlier stages of pathogenesis than in the case of CRP [77,78]. This may explain the results we obtained: a more significant difference in the levels of PCT than CRP as the 60-day model of feeding with a cholesterol diet and administration of Cornelian cherry extract can be considered relatively short when considering the development of chronic inflammatory diseases such as atherosclerosis, and it can be discussed as a limitation of the experiment. However, to our knowledge, this is the first study to assess the effect of *Cornus*-derived products on PCT levels, so despite its limitations, it is certainly an important step forward in broadening the knowledge in this aspect.

Another examination constraint worth mentioning is the application of two doses of the extract, which in the case of assessing as many parameters as there were in the described study, limits to a certain extent the possibility of a proper dose-dependence determination in comparison to, for example, using three doses. On the other hand, the strength of our study is that we assessed the impact of Cornelian cherry extract on the parameters for which there are relatively few reports or no reports at all. In addition, we did not observe any side effects during the experiment, which, combined with a relatively large number of significant changes in the levels of parameters evaluated, presents Cornelian cherry extract as a promising therapeutic alternative in the prevention and treatment of atherogenesis-related diseases.

## 4. Materials and Methods

### 4.1. Animal Model

In the current study, we used biobank blood and liver samples that were collected during the original experiment described in our previous article [27]. A total of 50 sexually mature male New Zealand rabbits aged 8 to 12 months were used in the 60-day experiment. The animals were housed in individual chambers with temperatures maintained at 21–23 °C. After four weeks of acclimatization and observation, the rabbits were randomly divided into 5 groups of 10 animals. The animals in group P were fed the standard chow for rabbits. Animals in other groups: CHOL, EXT 10, EXT 50, and SIMV 5 were fed with the standard chow enriched with 1% cholesterol. During the experiment, rabbits had free access to drinking water and received the same daily portion of chow (40 g/kg). Once daily, in the morning, for the consecutive 60 days of the study, the following substances were administered orally to the rabbits: groups P and CHOL—normal saline solution, group EXT 10—*Cornus mas* L. extract 10 mg per kg bw, group EXT 50—*Cornus mas* L. extract 50 mg per kg bw, and group SIMV 5—simvastatin 5 mg per kg bw as a positive control. The feeding schema is presented in Table 3.

Blood samples were taken from each animal at the beginning and the end of the experiment from the marginal vein of the ear or the saphenous vein. At the end of the study, the rabbits were put under terminal anesthesia. The aortas were harvested afterward and cleaned, then frozen and stored at −70 °C pending further analysis.

### 4.2. Plant Materials and Preparation of Extract

The research material was resin-purified Cornelian cherry fruit extract (*Cornus mas* L.). Fruits were collected at the Arboretum and the Institute of Physiography in Bolestraszyce, Poland. Before analysis, fruits were stored at −20 °C. The herbarium specimen (BDPA 3967) was authenticated and deposited at the Herbarium of the Arboretum and the Institute of Physiography in Bolestraszyce, Poland.

Frozen ripe *Cornus mas* L. fruits were ground and heated for 5 min at 95 °C with Thermomix (Vorwerk, Wuppertal, Germany). The pulp was cooled to 40 °C and depectinized at 50 °C for 2 h by adding 0.5 mL of Panzym Be XXL (Begerow GmbH & Co., Darmstadt, Germany) per 1 kg. After depectinization and pitting, the pulp was pressed in a laboratory hydraulic press (Zodiak, SRSE, Warsaw, Poland). The pressed juice was then filtered and passed through an Amberlite XAD-16 resin column (Rohm and Haas, Chauny Cedex, France). The impurities were rinsed with distilled water, while the pigments and iridoids were eluted with 80% ethanol. The eluate was concentrated under a vacuum at 40 °C. The solvent was evaporated using Rotavapor (Unipan, Warsaw, Poland). The concentrated dye and iridoid extract was purified with ethyl acetate to remove nonpolar impurities and other flavonoids. The purification procedure was repeated three times. A sample of the purified compounds was concentrated under a vacuum at 40 °C and lyophilized (Alpha 1-4 LSC, Christ, Germany) [79].

The main active ingredients of the obtained extract were iridoids, anthocyanins, phenolic acids, and flavonols. The quantitative composition of compounds in the EXT was determined by the HPLC-PDA method [80] and presented in our previous study [27].

### 4.3. RNA Isolation, Reverse Transcription, and Assessment of mRNA Expression of MMP-1, MMP-9, IL-6, NOX, and VCAM-1 in the Aorta by Real-Time PCR

Total RNA was isolated from studied tissue samples with RNeasy Fibrous Mini Kit (Qiagen, Hilden, Germany) according to the manufacturer’s protocol. To eliminate genomic DNA contamination, on-column DNase digestion was performed using RNase-Free DNase Set (Qiagen, Hilden, Germany). Quantity and purity of RNA samples were assessed by measuring the absorbance at 260 and 280 nm with a NanoDrop1000 spectrophotometer (Thermo Fisher Scientific, Wilmington, DE, USA). First-strand cDNA was synthesized using the High Capacity cDNA Reverse Transcription Kit (Applied Biosystems, Carlsbad, CA, USA) as described in the protocol. The mRNA expression of MMP-1, MMP-9, IL-6, NOX, and VCAM-1 was determined by quantitative real-time PCR with 7500 Real-Time PCR System and Power SYBR Green PCR Master Mix (Applied Biosystems, Carlsbad, CA, USA). Glyceraldehyde 3-phosphate dehydrogenase (GAPDH) was used as the reference gene. The reactions were performed with RT² qPCR Primer Assays (Qiagen, Hilden, Germany) for rabbit MMP-1 (PPN00411A), MMP-9 (PPN00307A), IL-6 (PPN00115A), NOX (PPN00200A), VCAM-1 (PPN00241A), and GAPDH (PPN00377A). All the reactions were performed in triplicates under the following conditions: activation of the polymerase at 50 °C for 2 min, initial denaturation at 94 °C for 10 min, and 40 cycles of denaturation at 94 °C for 15 s followed by annealing and elongation at 60 °C for 1 min. The specificity of the PCR was determined by melt–curve analysis for each reaction. The relative mRNA expression of the examined factors was calculated with the ∆∆Ct method.

### 4.4. Quantification of Serum Levels of VCAM-1, ICAM-1, CRP, PON-1, MCP-1, and PCT by Enzyme-Linked Immunosorbent Assay (ELISA)

ELISA method was used in the evaluation of serum levels of VCAM-1 (Rabbit VCAM-1 ELISA kit, CSB-E10092Rb, Cusabio Technology LLC, Houston, TX, USA), ICAM-1 (ELISA kit for Rabbit ICAM-1, ERB0114, Fine Test, Wuhan Fine Biotech Corp., Wuhan, China), CRP (C Reactive Protein Rabbit ELISA Kit, AB157726-1X, Abcam, Cambridge, UK), PON-1 (Elisa kit PON1 Rabbit, E0011Rb, Bioassay Technology Laboratory, Shanghai, China), MCP-1 (ELISA kit for Rabbit MCP-1, ERB0074, Fine Test, Wuhan Fine Biotech Corp., Wuhan, China) and PCT (ELISA kit for Rabbit PCT, ERB0144, Fine Test, Wuhan Fine Biotech Corp., Wuhan, China). All tests were performed according to the manufacturer’s instructions. All concentrations were expressed as ng/mL or pg/mL.

### 4.5. Statistical Analysis

Parametric data were expressed as mean ± standard deviation (mean ± SD). The statistical analysis was conducted using Statistica v. 13.3 software (TIBCO Software Inc., Palo Alto, CA, USA). One-way analysis of variance (ANOVA) with least significant difference (LSD) Fisher’s post hoc test was performed for a comparison between multiple groups. The *p*-values < 0.05 were considered statistically significant. Graphical representations of the statistical data were created using the Statistica v. 13.3 software (TIBCO Software Inc., Palo Alto, CA, USA).

## 5. Conclusions

The application of the iridoid and anthocyanin-rich resin-purified Cornelian cherry extract in a cholesterol-rich diet rabbit model resulted in a significant reduction in MMP-1, IL-6, and NOX mRNA expression in the aorta and a decrease in VCAM-1, ICAM-1, PON-1, MCP-1, and PCT serum levels, especially in the case of the 50 mg/kg bw dose. The results obtained, in conjunction with no observed side effects, indicate the potential usefulness of the Cornelian cherry extract either in the prevention or treatment of atherogenesis-related cardiovascular diseases, such as atherosclerosis or metabolic syndrome.

## Figures and Tables

**Figure 1 ijms-24-03890-f001:**
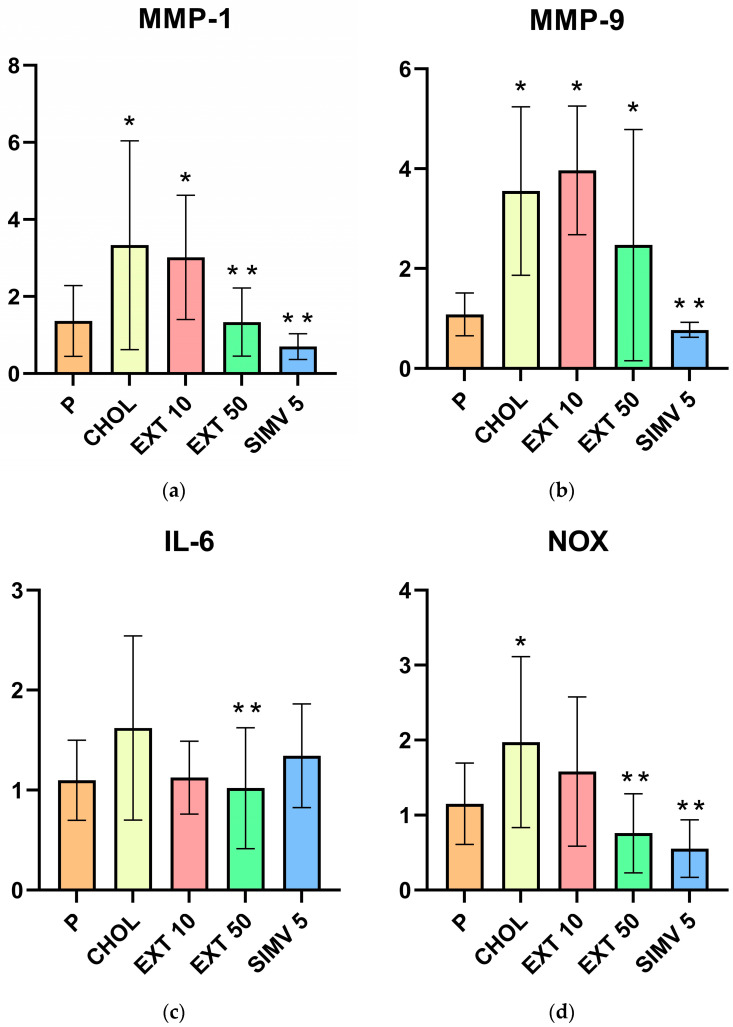
mRNA expression of MMP-1, MMP-9, IL-6, NOX, and VCAM-1 in the aorta. (**a**) MMP-1, (**b**) MMP-9, (**c**) IL-6, (**d**) NOX, and (**e**) VCAM-1. P—standard chow; CHOL—standard chow + 1% cholesterol; EXT 10—standard chow + 1% cholesterol + Cornelian cherry extract 10 mg/kg bw; EXT 50—standard chow + 1% cholesterol + Cornelian cherry extract 50 mg/kg bw; SIMV 5—standard chow + 1% cholesterol + simvastatin 5 mg/kg bw Values are presented as mean ± SD. * *p* < 0.05 vs. P. ** *p* < 0.05 vs. CHOL.

**Figure 2 ijms-24-03890-f002:**
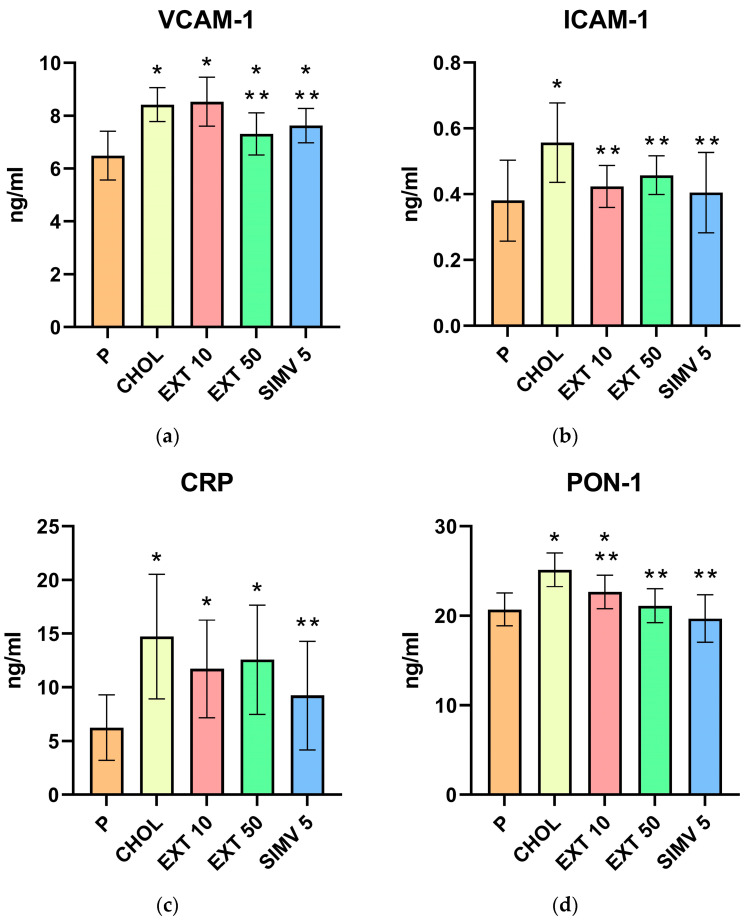
Serum levels of VCAM-1, ICAM-1, CRP, PON-1, MCP-1, and PCT by ELISA method. (**a**) VCAM-1, (**b**) ICAM-1, (**c**) CRP, (**d**) PON-1, (**e**) MCP-1, and (**f**) PCT. P—standard chow; CHOL—standard chow + 1% cholesterol; EXT 10—standard chow + 1% cholesterol + Cornelian cherry extract 10 mg/kg bw; EXT 50—standard chow + 1% cholesterol + Cornelian cherry extract 50 mg/kg bw; SIMV 5—standard chow + 1% cholesterol + simvastatin 5 mg/kg bw Values are presented as mean ± SD. * *p* < 0.05 vs. P. ** *p* < 0.05 vs. CHOL.

**Table 1 ijms-24-03890-t001:** mRNA expression of MMP-1, MMP-9, IL-6, NOX, and VCAM-1 in the aorta. Values are presented as mean ± SD.

Group	MMP-1	MMP-9	IL-6	NOX	VCAM-1
P	1.365 ± 0.918	1.082 ± 0.430	1.099 ± 0.400	1.153 ± 0.543	1.076 ± 0.429
CHOL	3.333 ± 2.709	3.556 ± 1.688	1.623 ± 0.921	1.971 ± 1.140	2.663 ± 0.787
EXT 10	3.017 ± 1.613	3.969 ± 1.287	1.125 ± 0.363	1.580 ± 0.995	2.807 ± 1.148
EXT 50	1.337 ± 0.885	2.470 ± 2.316	1.021 ± 0.606	0.757 ± 0.528	2.122 ± 1.872
SIMV 5	0.702 ± 0.335	0.773 ± 0.153	1.344 ± 0.518	0.553 ± 0.384	0.662 ± 0.266

**Table 2 ijms-24-03890-t002:** Serum levels of VCAM-1, ICAM-1, CRP, PON-1, MCP-1, and PCT by ELISA method. Values are presented as mean ± SD. VCAM-1, ICAM-1, CRP, PON-1, and PCT concentrations are expressed as ng/mL and MCP-1 as pg/mL.

Group	VCAM-1	ICAM-1	CRP	PON-1	MCP-1	PCT
P	6.490 ± 0.923	0.381 ± 0.123	6.241 ± 3.039	20.700 ± 1.828	137.949 ± 34.880	0.750 ± 0.331
CHOL	8.421 ± 0.644	0.557 ± 0.121	14.734 ± 5.797	25.135 ± 1.868	180.805 ± 51.942	1.585 ± 0.524
EXT 10	8.532 ± 0.931	0.424 ± 0.064	11.720 ± 4.535	22.660 ± 1.880	133.751 ± 36.279	1.358 ± 0.649
EXT 50	7.310 ± 0.801	0.458 ± 0.059	12.568 ± 5.080	21.114 ± 1.904	150.531 ± 49.552	1.068 ± 0.665
SIMV 5	7.630 ± 0.648	0.405 ± 0.122	9.239 ± 5.052	19.684 ± 2.643	132.770 ± 41.303	0.850 ± 0.539

**Table 3 ijms-24-03890-t003:** Experimental groups and the feeding schema used in the experiment.

Group	Chow	Dose of Tested Substance
P	standard chow	none (normal saline solution)
CHOL	standard chow +1% cholesterol	none (normal saline solution)
EXT 10	standard chow +1% cholesterol	Cornelian cherry extract 10 mg/kg bw
EXT 50	standard chow +1% cholesterol	Cornelian cherry extract 50 mg/kg bw
SIMV 5	standard chow +1% cholesterol	simvastatin 5 mg/kg bw

## Data Availability

The data underlying this article will be shared upon request with the corresponding authors.

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
