# Peer review of "Cornelian Cherry (Cornus mas L.) Iridoid and Anthocyanin-Rich Extract Reduces Various Oxidation, Inflammation, and Adhesion Markers in a Cholesterol-Rich Diet Rabbit Model"

_ijms, 2023, doi:10.3390/ijms24043890_

Round 1

Reviewer 1 Report

Danielewski et. al., determined the effect of of iridoid and anthocyanin-rich resin-purified cornelian cherry extract on markers important in the progress of inflammation and adhesion, immune system cells infiltration and atherosclerotic lesion development in cholesterol-rich diet rabbit model. They showed that cornelian cherry extract at a dose of 50 mg/kg b.w. resulted in a significant reduction of MMP-1, IL-6, and NOX mRNA expression in the aorta and a decrease of VCAM-1, ICAM-1, PON-1, and PCT serum levels. As they are nutraceuticals, they have a potential usefulness o in the prevention or in the treatment of atherogenesis-related cardiovascular diseases, such as atherosclerosis or metabolic syndrome. The study is important. However, I have some queries regarding the study.

Major point:

It is not clear whether they have used the atherosclerosis model in this study.

Whether the model used shown some symptoms of the development of atherosclerosis (for example)? Whether the application of cornelian cherry extract relieves these symptoms?

Authors are advised to represent the result for each group with individual values and use proper p-value representation. This is required for both the figures

Author Response

Respected Reviewer,

On behalf of the co-authors and ours, we would like to thank you for your time devoted to the thorough study of the content of our manuscript and all valuable hints and accurate comments. We are confident that they have helped us improve the manuscript markedly.

We provide our response to each suggestion below:

It is not clear whether they have used the atherosclerosis model in this study. Whether the model used shown some symptoms of the development of atherosclerosis (for example)? Whether the application of cornelian cherry extract relieves these symptoms?

 The model used in this manuscript was described in our earlier work, which we refer to several times in the text and which this manuscript is a continuation. The model used indeed showed the symptoms of atherosclerosis development, which was manifested, i.a., in increased intima and media thickness in the aorta, and cornelian cherry extract reduced these symptoms. We inform readers about this directly by writing in the introduction: “our previous work proved that cornelian cherry extract is an efficacious therapeutic agent in a cholesterol-rich diet rabbit model. We have shown, among others, the positive impact of the extract on the levels of triglycerides and adipokines, PPAR alpha and gamma expression in the aorta, and LXR expression in the liver, as well as on the reduction of the intima/media ratio in the thoracic and abdominal aorta”.

Authors are advised to represent the result for each group with individual values and use proper p-value representation. This is required for both the figures.

It’s a very accurate comment. We extended the content with tables showing values for the surveyed groups and corrected the p-value representation.

In addition, the manuscript was proofread in the “Grammarly” application and reviewed by a native speaker, which we hope will improve the reception of our work.

We hope you will be satisfied with the revisions and re-consider our manuscript for publication in the IJMS journal. Thank you very much.

Kind regards

Maciej Danielewski & Tomasz Sozański

Reviewer 2 Report

-

Author Response

Respected Reviewer,

On behalf of the co-authors and ours, we would like to thank you for your time devoted to the thorough study of the content of our manuscript. We are very grateful for your positive evaluation.

Kind regards

Maciej Danielewski & Tomasz Sozański

Reviewer 3 Report

In this study, the authors determined the effect of iridoid and anthocyanin-rich resin-purified cornelian cherry extract on inflammation, cell proliferation, adhesion, immune system cells infiltration, and atherosclerotic lesion development in cholesterol-rich diet rabbit model. Although there are some interesting results in this study, but the novelty and quality are not satisfied with the requirement of IJMS journal.

These are some comments:

(1) The language should be carefully revised all through this manuscript.

(2) The “Introduction” and “Discussion” section were too long and not focus, the authors should rewrite these to section and make them more concise and readable.

(3) The Figures should be revised carefully, these forms of figures were not commonly used in scientific papers, such as the symbols used to present the difference between groups were appropriate, we often use “*, **, ***” to present difference.

(4) The data in this study were not enough, more experiments should be carried out to make the results more intact and persuasive.

Author Response

Respected Reviewer,

On behalf of the co-authors and ours, we would like to thank you for your time devoted to the thorough study of the content of our manuscript and all valuable hints and accurate comments. We are confident that they have helped us improve the manuscript markedly.

We provide our response to each suggestion below:

The language should be carefully revised all through this manuscript.

The manuscript was proofread in the “Grammarly” application and reviewed by a native speaker, which we hope will improve the reception of our work.

The “Introduction” and “Discussion” section were too long and not focus, the authors should rewrite these to section and make them more concise and readable.

Thank you very much for this comment. We shortened both sections trying to make them more concise and readable.

The Figures should be revised carefully, these forms of figures were not commonly used in scientific papers, such as the symbols used to present the difference between groups were appropriate, we often use “*, **, ***” to present difference.

It’s a very accurate comment. We corrected the p-value representation and extended the content with tables showing values for the surveyed groups.

The data in this study were not enough, more experiments should be carried out to make the results more intact and persuasive.

The model used in this manuscript was described in our earlier work, which we refer to several times in the text and which this manuscript is a continuation and supplement. The experiment has already been completed - we have no means, including financial ones, to extend the data obtained. However, we believe that the results described in this manuscript are a significant complement to the previous report. In addition, for most parameters, such as MMP-1, MMP-9, VCAM-1, ICAM-1, IL-6, CRP, and PON-1, our manuscript is one of the few describing the effect of the product obtained from Cornus Mas, and in the case of PCT, to our knowledge, our manuscript is the first. Therefore, we hope this manuscript is a valuable addition and extension of knowledge about the possible therapeutic applications of cornelian cherry.

We hope you will be satisfied with the revisions and re-consider our manuscript for publication in the IJMS journal. Thank you very much.

Kind regards

Maciej Danielewski & Tomasz Sozański

Round 2

Reviewer 1 Report

I appreciate the author's effort to incorporate the suggestions made by me and am satisfied with the changes made in the manuscript.

Author Response

Respected Reviewer,

Thank you kindly for your valuable hints. We appreciate your cooperation very much.

Kind regards

Maciej Danielewski & Tomasz Sozański
